# Effect of Titanium-Base Abutment Height on Optical Properties of Monolithic Anterior Zirconia Crowns

**DOI:** 10.3390/ma15217643

**Published:** 2022-10-31

**Authors:** Ameer Biadsee, Mutlu Özcan, Lubaba Masarwa, Mishel Haddad, Nadin Al-Haj Husain, Zeev Ormianer

**Affiliations:** 1Department of Oral Rehabilitation, The Maurice and Gabriela Goldschleger School of Dental Medicine, Tel Aviv University, Tel Aviv 69978, Israel; lubabam@mail.tau.ac.il (L.M.); mishelhaddad@mail.tau.ac.il (M.H.); ormianer@tauex.tau.ac.il (Z.O.); 2Division of Dental Biomaterials, Clinic of Reconstructive Dentistry, Center of Dental Medicine, University of Zurich, 8032 Zurich, Switzerland; mutlu.ozcan@zzm.uzh.ch (M.Ö.); nadin.al-hajhusain@zzm.uzh.ch (N.A.-H.H.); 3Department of Reconstructive Dentistry and Gerodontology, School of Dental Medicine, University of Bern, 3010 Bern, Switzerland

**Keywords:** colour, dental, dental materials, monolithic zirconia, optical properties, prosthetic dentistry, titanium-base abutment

## Abstract

The effects of different heights of ti-base abutments on the color of anterior screw-retained zirconia restorations fabricated using computer-aided design and computer-aided manufacturing (CAD-CAM) technologies may affect the optical clinical outcome. The purpose of this study was to measure and compare the color parameters of zirconia crowns in different shades on ti-base abutments. Identical specimens (N = 160) were milled to restore the screw-retained central maxillary incisor crown, using 5% mol yttria zirconia (5Y-TZP). The specimens were designed using computer design software to match 3.5 mm and 5.5-mm ti-base abutments and milled using one CAD-CAM technology. Specimens were divided into four main groups depending on zirconia shade (A1/0, A2/3, A3.5/4 and B2/3) and then assigned to two subgroups according to ti-base height. Color measurements in the CIELab coordinates were made using a spectrophotometer under room-light conditions. Color difference (ΔE*) values were calculated using the CIE76 and CIEDE2000 formula. Within the group of A0/1 and A2/3, for 5.5 mm abutment height, a significant difference was found between the means of colors ∆E00 and ∆Eab (*p* < 0.01). Using a 5.5 mm-height ti-base abutment may produce a clinically unacceptable outcome (ΔEab > 2) in A1/0 and A2/3 color groups.

## 1. Introduction

An anterior fixed prosthesis aims to mimic nature by providing the same characteristics for natural teeth as texture, size, shape and color properties. Implant-supported restoration can be applied by two different techniques, either a screw-retained prosthesis or a cement-retained prosthesis [1]. The use of prefabricated titanium-base abutments (ti-base) for two-piece zirconia abutments are widely used mainly due to lower fracture rates compared to a one-piece zirconia abutment [2]. Increasing the production of monolithic lithium disilicate or zirconia using CAD-CAM also led to a wide use of ti-base abutments [3,4,5]. Different ti-base design and heights are presented by manufacturers for distinct clinical indications. Cementing the abutments extra-orally should reduce the risk of cement-induced peri-implantitis [6]. However, these abutments lack translucency and could affect light scattering through the zirconia and thereby the aesthetics of the restoration.

One recent study evaluated the effect of ti-base abutment height on the retention of zirconia crowns using a pull-out test and concluded that the ti-base height has no effect on zirconia superstructure retentiveness [7]. However, many clinicians and dental technicians prefer to use a higher ti-base abutment whenever occlusal-gingival space permits, without taking into account the aesthetic consequences.

Quantifying tooth color is usually performed via the CIELab system, using the computerized data obtained by a spectrophotometer, which allows for mathematical analysis and comparison of color properties [8,9,10]. According to the Commission Internationale de l’Eclairage (CIE) Lab color coordinates, L* represents the lightness of the object, a* represents the location of the object on the blue/green to red/purple axis, and b* represents the location of the object on the purple/blue to yellow axis [11]. Color change (ΔE*) could be assessed by calculating the L (Value)*a (Chroma)*b (Hue)* equation [11,12]. Complete contour crown thickness and the zirconia’s disk translucency may affect these parameters, similar to other ceramics [13]. Different studies have tried to evaluate the minimal thickness of the ceramic needed to mask a dark substrate [14]; as reported for other ceramics, the thickness of the zirconia may affect the color-masking ability. Studies have suggested a minimal thickness of 1–2 mm of translucent zirconia [15]. An attempt to overcome the aesthetic challenges using an opaque cement material may result in unpleasant aesthetic results [13,16].

Previous studies have reported a variability in the findings of the color threshold using the CIE76 formula [8,9,10], ΔE* = 1 of the 50:50 perceptibility and a ΔE* of 2 up to 3.7 of the 50:50 acceptability [17]. In other words, only half of observers will notice a color mismatch above ΔE* = 1, and half of the observers will report an unacceptable color above ΔE* = 2.16.

Few studies, however, have examined the effect of several factors on the light scattering of zirconia, such as material composition, porosity, sintering and grain size, concluding that reducing oxide alumina particles (Al_2_O_3_), minimizing pores sizes, evaluating sintering temperatures and using larger grains can enhance the monolithic zirconia esthetic properties [18].

Although ti-base abutment height may have an influence on zirconia superstructures’ retentiveness and light scattering, limited data are available investigating the influence on anterior zirconia restorations’ translucency and aesthetics. The purpose of the present study was to evaluate the effect of ti-base abutment height on the color of different translucent anterior zirconia crown shades. The null hypothesis was that the ti-base abutment height would not affect the final color of different zirconia in different shades.

## 2. Material and Methods

An implant scan-body (RC Scan-body; Straumann, Basel, Switzerland) was screwed to an implant analog (RC analog; Straumann, Basel, Switzerland) embedded in acrylic resin (Unifast3; GC Corporate, Tokyo, Japan), then scanned using a laboratory scanner (D700; 3shape, Copenhagen, Denmark). The 3D digital data was inserted into the computer software (DentalCAD V2.3, Matera, Exocad) in order to virtually design two identical monolithic screw-retained zirconia implant central incisor crowns to match 3.5 mm and 5.5 mm ti-base abutments (RC Variobase for crown, Gingival Height 1 mm, Straumann, Basel, Switzerland). The crown was designed with a facial aspect thickness of 0.8 mm at the ti-base finish line up to 2 mm at the mid-occlusal-gingival crown height and with screw-channel located in the mid-buccal-palatal aspect (Figure 1).

The monolithic zirconia crowns were milled from a 5 mol% yttria-stabilized tetragonal zirconia polycrystal (5Y-TZP) block (Zolid Fx multilayer; Amann Girrbach, Koblach, Austria) using CAD-CAM technology (Ceramil Motion2; Amann Girrbach) in a dry fine milling strategy. Three cutter tools: 2.5 mm, 1 mm and 0.6 mm, were used. Each set of 3 new cutter tools was for milling 90 specimens. In order to mimic clinical conditions, the crowns were hand polished using diamond polish mint paste (Ultradent, Tokyo, Japan) and glazed using a synthetic disc (Disc Buff, Shofu Inc., Tokyo, Japan) with porcelain polishing paste (Pearl Surface Z; Kuraray, Tokyo, Japan) in a single direction for 30 s by the same operator (AB).

One hundred and sixty CAD-CAM ti-base abutments with two heights of 3.5 and 5.5-mm (RC Variobase; Straumann) were screwed to resin-embedded implant analogs (RC implant analog; Straumann) (Figure 2).

Specimens were divided into 4 main groups depending on zirconia shade, then divided to 2 subgroups according to ti-base height; no cement was used to prevent cement color bias (Table 1).

A digital spectrophotometer (VITA Easyshade; VITA Zahnwfabrik, Bad Säckingen, Germany) was used for color measurements. For the color measurement, a special mold was used to standardize the specimen’s position, making the spectrophotometer probe tip axis perpendicular to the tested specimen’s buccal surface. The device was calibrated according to the manufacturer’s instructions using a calibration apparatus before each subgroup, and all measurements were made in the optical environment. All the specimens were tested with a standardized-photography neutral 18% gray card (Kodak Gray Cards; Tiffen Co., Kingston, Australia) as a background. Room illuminance was 1007 lux, which was measured using a light meter (Testo 540 Lux Meter; Testo, Titisee, Germany). Specimens were randomized in each group, and the spectrophotometric measurements were performed and recorded by a single operator (ML) with the Commission International de I’Eclairage (CIE) L*a*b* color space system, which allows for the determination of color in the three-dimentional space. L* indicates the lightness (0–100), with 0 being black and 100 being white. The coordinate a* is for red and green, and b* is for yellow and blue.

The CIE76 and the CIEDE2000 formulas based on the CIELAB color space are as follows:
(1)
$$\mathrm{CIE}76=\Delta E_{\mathrm{ab}}=\sqrt{a^{2}+b2+b^{2}}$$


(2)
$$\mathrm{CIEDE}2000=\Delta E_{00}=[(\frac{\Delta L^{\prime }}{\mathrm{kLSL}}{)}^{2}+(\frac{\Delta C^{\prime }}{\mathrm{kCSC}}{)}^{2}+(\frac{\Delta H^{\prime }}{\mathrm{kHSH}}{)}^{2}+R_{T}\;(\frac{\Delta C^{\prime }}{\mathrm{kCSC}})(\frac{\Delta H^{\prime }}{\mathrm{kHSH}})]\frac{1}{2}$$


In the present study, the parametric factors of the CIEDE2000 formula were set to 1. Further, the perceptibility threshold was set at ΔE_00_ ≤ 1.30 and Δ*_ab_* ≤ 1.00. Clinical acceptability was set at ΔE_00_ > 2.25 and Δ*_ab_* > 2 [7].

Data analysis was performed using SPSS 27 for Windows (IBM SPSS Statistics, Chicago, IL, USA). Descriptive analysis included mean, median and standard deviation. The Kolmogorov–Smirnov Test showed a non-normal distribution, and an E log transformation was made. Data were analyzed using two-way ANOVA, and mean ΔE values were compared with a *t*-test (alpha = 0.05 for all tests).

To test for the difference in mean between the 2 groups, the *t*-test was used according to a sample size calculation that considers test significance = 0.05, power = 0.9 and effect size = 1.6.

(Effect size assumed a difference of 1.2 between the 2 groups, with a pooled standard deviation of 0.75). As performed in SPSS software, version 21, 9 samples in each group should be sufficient for achieving significant results at 0.05.

## 3. Results

The means and standard deviations of the color parameters L*, a* and b* of test groups are presented in Table 2.

The A3/4 color group had the lowest L* variable values, (−2 ± 0.73) for a 3.5-mm abutment height and (−2.02 ± 0.54) for a 5.5-mm abutment height. The B2/3 color group had the highest b* variable values, (1.03 ± 0.59) for a 3.5-mm abutment height and (1.72 ± 1.12) for a 5.5 abutment height. ∆E* values were calculated using the CIE76 and CIEDE2000 formula. Two-way ANOVA for the independent variables, abutment height and its interaction with crown color significantly influenced the ΔE values in the study groups (*p* < 0.01) (Table 3).

Means and standard deviations of the CIE76 and CIEDE2000 color difference of all groups are presented in Table 4. Box plots of the minimum, maximum, interquartile range, medians and the outliers for each of the L*, a* and b* variables are presented in Figure 3a–c.

## 4. Discussion

The results of the present study suggest that the abutment height significantly affects the Δa*, Δb*, ΔL* and ΔE* variables of different zirconia crown shades. Thus, the null hypothesis was rejected.

There are two main methods for color evaluation: visual and instrumental and commercially manufactured shade guides are used for the visual method, which is extremely difficult, due to multiple factors, such as environmental and lightening conditions, as well as viewer interpretation. In an objective color analysis with the spectrophotometer used in this study, according to other previous studies, more reliable and accurate results were found than subjective methods [17,19]. Previous literature mentioned that the value of ΔE = 1 is visually detectable 50% of the time and may be noticed by some observers [17,18]. For the acceptability threshold, values ranged between 2 and 3.7 [17,18].

In our study, ΔE* values using the CIE76 formula ranged from 0.96 to 2.53, with the highest value (ΔE_ab_ = 2.53) found in the A1/0 color group and a value of (ΔE_ab_ = 2.23) in the A2/3 color group when using a 5.5-mm abutment height. These findings, which were approved by different previous studies, are considered clinically unacceptable.

Similarly, ∆E* values using the CIEDE2000 formula were the highest in groups A1/0 and A2/3 and ranged from 1.04 to 2.47. ∆E* values calculated using CIEDE2000 were mostly higher than values calculated using CIE76. On the other hand, these results must be interpreted with caution, taking into account that this is an in vitro study and that clinical trials with patient-reported outcomes are needed in order to verify the clinical importance of these results.

One previous study reported the effect of abutment color on the restoration color and suggested increasing the thickness of the restoration when using a titanium abutment [14]. This solution may be not acceptable when restoring using a translucent monolithic anterior crown; increasing the thickness of the anterior crown may be not possible due to neighboring restoration colors and harmony. As reported in previous studies, trying to overcome this challenge using an opaque cementing material may result in an unacceptable outcome [13,15,16]. One previous in vitro study investigated the effect of cement shade on the color of a translucent zirconia 1 mm disk and found a value of (ΔE_ab_ = 2.59 ± 0.05) when using a translucent-shade cement before and after the cementation procedure, which may be considered clinically unacceptable. Moreover, an opaque cement results in a clinically unaccepted color change in 0.5 mm and in 2 mm zirconia disk thicknesses. Hence, to prevent bias, ti-base abutments were not cemented in this study [13]. One other study suggested that a 1.5-mm zirconia thickness is sufficient to achieve optimal masking results [16]. L* values were significantly different in the A1/0 and B2/3 color groups compared to the control, meaning that tested specimens produced different brightness. Moreover, a* values were significantly different in the A1/0 color group: using a 5.5 mm ti-base resulted in a more greenish result. On the other hand, b* values were significantly higher in the A1/0 and A2/3 color groups: using a 5.5 mm ti-base resulted in more blueish results. There was no significant statistical difference in the A3/4 color group for any color variable; this can be explained by the low translucency of the specimen, resulting in masking of the abutment color.

To achieve higher translucency, attempts were made by producing partially stabilized 4Y- and 5Y-TZP materials with a high nonbirefringent cubic phase content. Nowadays, we know that the translucency of zirconia materials increases with higher Y-TZP contents (5Y-TZP > 4Y-TZP > 3-YTZP). Zirconia with a microstructure with a grain size of under 100 nm is expected to allow for light transmission without scattering. Therefore, speed and super-speed sintering methods resulting in ultrafine and dense Y-TZP grains by growth prevention are used to achieve high translucency [20]. Therefore, if 5Y-TZP zirconia materials are applied in the aesthtic region using a ti-base, speed- and super-speed sintering methods should be avoided.

For more healthy soft tissues in the aesthetic zone, the crown’s ti-base interface is usually located 1 mm sub-gingivally, and different gingival ti-base heights can be used depending on the depth of the implant.

Silva et al. [7] reported that ti-base height did not affect the zirconia superstructures’ retentiveness; however, the cementation procedure did not include aluminum oxide sandblasting or treating the zirconia surfaces with 10-methacryloxydecyl dihydrogen phosphate primer, which may have a beneficial effect on retentiveness.

The limitations of this in vitro study include the in vitro color assessment, the fact that only one thickness of restoration was tested, and the single translucent zirconia brand. These factors may lead to different spectrophotometric results. Another limitation of this is study is the that cement effect was not tested when each specimen acted as its own control group. The results of this study may, however, help setting guidelines for anterior zirconia implant reconstructions.

## 5. Conclusions

From this study, the following conclusions were drawn:Using a 5.5 mm-height ti-base abutment may produce a clinically unacceptable outcome (ΔE_ab_ > 2) in A1/0 and A2/3 shades of monolithic translucent zirconia.When restoring A1/0 and A2/3 shades of monolithic anterior zirconia restorations using high ti-base abutments, the final color may be largely affected.If possible, using a short ti-base abutment may be favorable when restoring a monolithic anterior A1/0 and A2/3 shade zirconia, providing that the retention forces must be considered, depending on the length of the restored crown.

## Figures and Tables

**Figure 1 materials-15-07643-f001:**
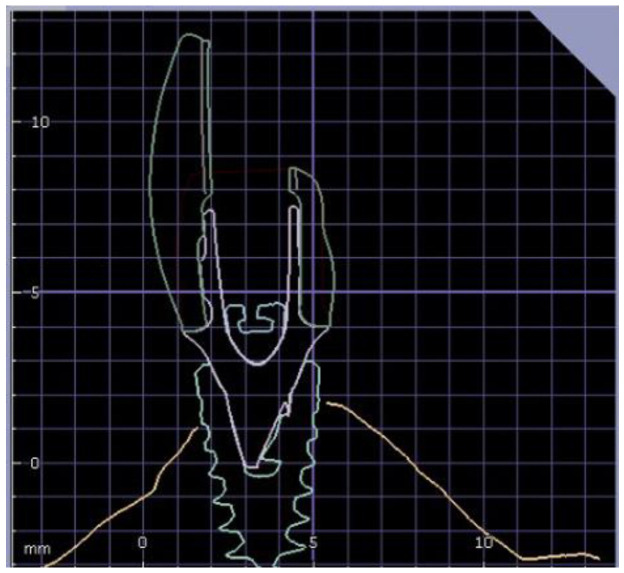
Crown design, showing facial thickness.

**Figure 2 materials-15-07643-f002:**
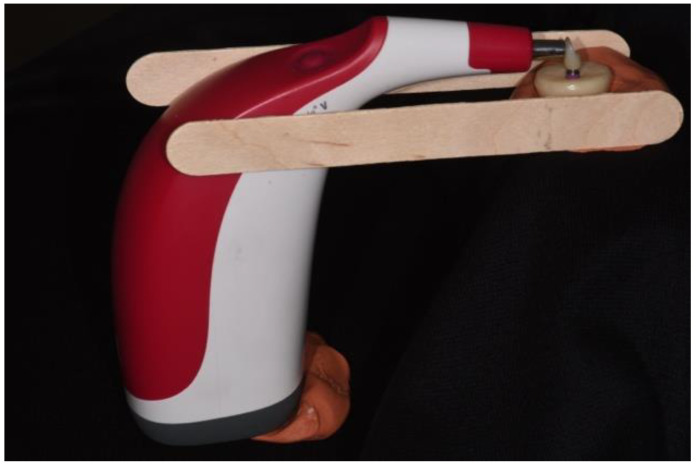
Implant analog embedded into acrylic resin with a screw-retained zirconia crown using ti-base abutment, and spectrophotometric testing was performed using a special mold to hold the device while touching the crown.

**Figure 3 materials-15-07643-f003:**
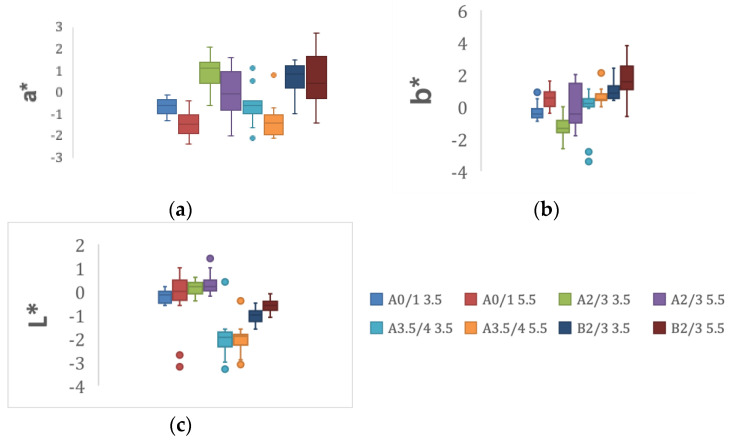
(**a**–**c**). Box plots of the minimum, maximum, interquartile range, medians and the outliers for each of the (**a**) a*, (**b**) b* and (**c**) L* variables in the study.

**Table 1 materials-15-07643-t001:** Grouping according to crown color and ti-base abutment height.

Group (*n* = 20)	Crown Color Shade	Ti-Base Abutment Height
A	A0/1	3.5
B		5.5
C	A2/3	3.5
D		5.5
E	A3.5/A4	3.5
F		5.5
G	B2/3	3.5
H		5.5

In each color group, the 20 specimens without the ti-base served as control group.

**Table 2 materials-15-07643-t002:** Means and standard deviations of the color parameter L*, a* and b* of test groups.

Color Group/Abutment Height	L*	a*	b*
A 0/1			
3.5 mm	−0.25 (0.27)	−0.64 (0.38)	−0.34 (0.45)
5.5 mm	−0.16 (1.05)	−1.38 (0.60)	0.47 (0.55)
A 2/3			
3.5 mm	0.12 (0.28)	0.88 (0.72)	−1.30 (0.63)
5.5 mm	0.29 (0.38)	0.04 (0.97)	0.09 (1.26)
A 3.5/4			
3.5 mm	−2.00 (0.73)	−0.59 (0.83)	−0.005 (1.11)
5.5 mm	−2.02 (0.54)	−1.36 (0.67)	0.60 (0.45)
B 2/3			
3.5 mm	−1.04 (0.32)	0.60 (0.77)	1.03 (0.59)
5.5 mm	−0.61 (0.23)	0.54 (1.13)	1.72 (1.12)

**Table 3 materials-15-07643-t003:** Two-way ANOVA for the independent variables, abutment height and its interaction with crown color significantly influenced the ΔE values in the study groups.

Dependent Variable:					
Source	Type III Sum of Squares	df	Mean Square	F	Sig.
Corrected Model	4.025 ^a^	7	0.575	3.349	0.002
Intercept	266.501	1	266.501	1551.878	0.000
COLOR	0.632	3	0.211	1.227	0.302
HT	1.398	1	1.398	8.142	0.005
COLOR * HT	1.995	3	0.665	3.873	0.011
Error	26.103	152	0.172		
Total	296.629	160			
Corrected Total	30.128	159			

^a^ R Squared = 0.134 (Adjusted R Squared = 0.094).

**Table 4 materials-15-07643-t004:** ΔL*, Δa*, Δb* and ΔE* between the combination of color group/abutment height and each control.

		ΔL*	Δa*	Δb*	ΔE CIE76*	ΔE CIEDE2000
Color group	Abutment height					
**A0/1**	3.5	0.22450	0.48050	0.47850	0.96 ± 0.51	1.04 ± 0.61
	5.5	2.02450	1.49700	2.25750	2.52 ± 0.65	2.18 ± 1.00
*p* value		0.01	0.005	0.002	0.001	<0.001
**A2/3**	3.5	0.37700	1.12500	1.01750	1.40 ± 0.75	1.57 ± 0.91
	5.5	0.70600	1.90900	3.19800	2.23 ± 0.92	2.47 ± 1.02
*p* value		0.155	0.074	0.003	0.003	0.004
**A3.5/4**	3.5	0.65200	1.24850	2.78950	1.82 ± 1.20	1.90 ± 1.28
	5.5	1.21400	2.00900	3.75850	2.04 ± 1.71	2.15 ± 1.77
*p* value		0.198	0.291	0.600	0.965	0.799
**B2/3**	3.5	0.25000	1.38150	2.94000	1.73 ± 1.27	1.91 ± 1.42
	5.5	0.07700	1.03550	1.13200	1.34 ± 0.66	1.58 ± 0.79
*p* value		0.004	0.578	0.096	0.393	0.465

## Data Availability

Data will be provided on request.

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
