# Peer review of "Effect of Titanium-Base Abutment Height on Optical Properties of Monolithic Anterior Zirconia Crowns"

_materials, 2022, doi:10.3390/ma15217643_

Round 1
Reviewer 1 Report
The discussion topic was very limited. In the introduction, 3 articles that are already used are cited. His work does not contain much other than its results. It should be supported by new citations and discussion comments.
In addition, very few studies are included in the experiment and/or simulation part. It can be made into a study in which other features of Ti-supported structures are also processed. There will be "short communication" only under the optical aesthetic title.
Author Response
Dear reviewer 1,
Thank you for your valuable comments.
work does not contain much other than its results. It should be supported by new citations and discussion comments.
In addition, very few studies are included in the experiment and/or simulation part. It can be made into a study in which other features of Ti-supported structures are also processed. There will be "short communication" only under the optical aesthetic title.
Answer: The manuscript was revised thoroughly; other studies were added.
Reviewer 2 Report
Dear authors,
Your paper investigates a topic of some current interest. The methodology seems fair but I have a few comments and suggestions.
The language is fair but the paper is in need of proof reading and could benefit further from consultation of a professional translator.
Introduction
Succinct and sufficient. Only minor issues to address.
I lack some comments regarding esthetics as this is a main topic of the research question. Perhaps mention that the use of monolithic zirconia is driven by esthetic demands and that the modern zirconia materials have increased translucency which makes the issue of the base an important influencing factor on final outcome?
Please rephrase the following on lines 34-35 “either a screw-retained 34 implant or cement-retained implant”. It is not the implant that is retained, it is the crown.
Lines 35-37. “The use of prefabricated titanium base abutment 35 (ti-base) is widely used due to the increasing production of monolithic lithium disilicate 36 or zirconia using CAD-CAM”. The use of Ti-bases to support zirconia crowns is mainly to avoid previous fractures of all-in-one zirconia abutments.
Lines 60-63 fit better as a comment in the discussion than in the intro. But that is just an opinion and suggestion.
Materials and methods
Clear and concise. No comments.
Discussion
Succinct discussion of the main finding. I have only one minor comment and one suggestion.
You use the words clinically acceptable/unacceptable. I would suggest adding a comment in the discussion that highlights the fact that this is an in vitro study and that clinical trials, preferably including patient-reported outcomes are needed in order to verify the “clinical” importance of the outcome of your study.
In the introduction, you mention the paper by Silva et al and that the height of abutments had no significant effect on outcome. But if I remember correctly, the larger platform diameters resulted in significantly better retention. This could be worth mentioning.
Line 158: “Two main methods for color evaluation, visual and instrumental.” This sentence seems unfinished.
Conclusions
No comments
Author Response
Dear reviewer 2,
Thank you for your valuable comments.
Dear authors,
Your paper investigates a topic of some current interest. The methodology seems fair but I have a few comments and suggestions.
The language is fair but the paper is in need of proof reading and could benefit further from consultation of a professional translator.
Introduction
Succinct and sufficient. Only minor issues to address.
Please rephrase the following on lines 34-35 “either a screw-retained 34 implant or cement-retained implant”. It is not the implant that is retained, it is the crown.
Answer: LINE 37-38, either a screw-retained prosthesis or cement-retained prosthesis [1].
Lines 35-37. “The use of prefabricated titanium base abutment 35 (ti-base) is widely used due to the increasing production of monolithic lithium disilicate 36 or zirconia using CAD-CAM”. The use of Ti-bases to support zirconia crowns is mainly to avoid previous fractures of all-in-one zirconia abutments.
Answer: LINE 38-41, The use of prefabricated titanium-base abutments (ti-base) for 2-piece zirconia abutments are widely used mainly due to a lower fracture rates compared to 1-piece zirconia abutment [2]. Increasing production of monolithic lithium disilicate or zirconia using CAD-CAM, also led to a wide use of ti-base abutments [3-5].
Lines 60-63 fit better as a comment in the discussion than in the intro. But that is just an opinion and suggestion.
Answer: the point was added to the discussion section.
Materials and methods
Clear and concise. No comments.
Discussion
Succinct discussion of the main finding. I have only one minor comment and one suggestion.
You use the words clinically acceptable/unacceptable. I would suggest adding a comment in the discussion that highlights the fact that this is an in vitro study and that clinical trials, preferably including patient-reported outcomes are needed in order to verify the “clinical” importance of the outcome of your study.
Answer: LINE 437-440, On the other hand, these results must be interpreted with caution, noting the fact that this is an in vitro study, and that clinical trials with patients-reported outcomes are needed in order to verify the clinical importance of these results.
In the introduction, you mention the paper by Silva et al and that the height of abutments had no significant effect on outcome. But if I remember correctly, the larger platform diameters resulted in significantly better retention. This could be worth mentioning.
Answer: In the study by Saliva et al, only one platform was used.
Line 158: “Two main methods for color evaluation, visual and instrumental.” This sentence seems unfinished.
Answer: LINE 409-412, There is two main methods for color evaluation, visual and instrumental and commercially manufactured shade guides are used for the visual method, which is extremely difficult, due to multiple factors, such as environmental and lightening conditions, and viewer interpretation
Conclusions
No comments
Thank you.
Reviewer 3 Report
You can find attached the PDF file.

Author Response
Dear reviewer 3,
Thank you for your valuable comments.
Abstract:
Answer: The abstract was revised.
Introduction:
Answer: The introduction section was revised.
Material and Methods:
The section was revised thoroughly.
Answer: LINE 195-199, The monolithic zirconia crowns were milled from a 5 mol% yttria-stabilized tetragonal zirconia polycrystal (5Y-TZP) block (Zolid Fx multilayer; Amann Girrbach) using CAD-CAM technology (Ceramil Motion2; Amann Girrbach) in dry fine milling strategy. Three cutter tools: 2.5 mm, 1 mm and 0.6 mm, were used. Each set of 3 new cutter tool was for milling 90 specimens. (according to manufacturer instructions).
LINE 228-229, making the spectrophotometer probe tip axis perpendicular to the tested specimen’s buccal surface.
LINE 232-234, Room illuminance was 1007 lux, which was measured using a light meter (Testo 540 Lux Meter; Testo). Specimens were randomized in each group, and the spectrophotometric measurements
- LINE 241-242, CIE76 = DEab =
- CIEDE2000 = ΔE00 = [()2+()2+()2+RT ()( )]
Results: All figure and tables are mentioned in the text. Legends added.
Discussion: revised.
Conclusions: revised.
Reviewer 4 Report
This is an interesting manuscript reporting an in vitro study aimed to measure and compare optical properties of zirconia in different shades on ti-base abutments. Some corrections, modifications, clarifications are needed to be publishable. These are provided below according to manuscript section:
Title:
- Authors should indicate that this is an “in vetro study” in the title.
- “CAD-CAM abutment” is misleading and can be mixed with custom-milled abutments. Kindly omit “CAD-CAM” from the title, or replace it with “Titanium-Base”.
Abstract:
- “The effects f computer-aided design”, I believe authors meant “the effects of…”.
- Kindly omit all the brands and trademarks from the abstract, only specify them in the methods section. E.g., (Zolid fx; Amann 17 Girrbach), (Motion2; Amann Girrbach), (DenatlCAD, V2.3, Matera; Exocad), (RC Variobase 21 for crown, Gingival height 1mm, Straumann), (VITA EasyShade; VITA Zahnfabrik), etc.
- “The purpose of this study is to measure and compare color dimensions of zirconia”. Please specify the type of zirconia restoration (e.g., zirconia crowns).
- Study groups were not clarified in the abstract. Define the study groups before presenting their results.
- Please add a clinical take-home message to the conclusions in the abstract.
Introduction:
- Replace “Anterior fixed partial dentures (FDP)” with “anterior fixed prosthesis” or “anterior fixed prosthodontic restorations”.
- The following is not accurate: “screw-retained implant or cement-retained implant”. I believe this was a mistranslation, and the author meant “implant-supported screw-retained and cement-retained prostheses”.
- Authors justified the reason of conducting this study.
Material and Methods:
- Please clarify how position was standardized while using Easyshade spectrophotometer probe. This is very important. A slight change in the position could result in different results. A clear illustrating figure could be very beneficial. Figure 2 did not show the special mold that was used to hold the Easyshade device clearly, although it was claimed to do so. Please explain how this was repeatable in all specimens.
- In sample size calculations, authors claimed that effect size was 1.6. Please explain where this came from. Sample size calculations should be explained in more details.
Results:
- Results are OK
Discussion:
- It is better to enrich the discussion section with more published results and comparisons.
- Please justify why ti-base abutments were chosen for the present study.
- Cite publications that confirm the validity of using Easyshade.
Conclusions:
- While drawing conclusions, kindly indicate that these were within the study limitations.
- Add clinical take-home messages to your conclusions
Author Response
Dear reviewer 4,
Thank you for your valuable comments.
This is an interesting manuscript reporting an in vitro study aimed to measure and compare optical properties of zirconia in different shades on ti-base abutments. Some corrections, modifications, clarifications are needed to be publishable. These are provided below according to manuscript section:
Title:
- Authors should indicate that this is an “in vetro study” in the title.
Answer: Effect of Titanium-Base abutment height on monolithic anterior zirconia crown optical properties- An In Vitro Study
- “CAD-CAM abutment” is misleading and can be mixed with custom-milled abutments. Kindly omit “CAD-CAM” from the title, or replace it with “Titanium-Base”.
Answer: revised
Abstract:
- “The effects f computer-aided design”, I believe authors meant “the effects of…”.
Answer: revised
- Kindly omit all the brands and trademarks from the abstract, only specify them in the methods section. E.g., (Zolid fx; Amann 17 Girrbach), (Motion2; Amann Girrbach), (DenatlCAD, V2.3, Matera; Exocad), (RC Variobase 21 for crown, Gingival height 1mm, Straumann), (VITA EasyShade; VITA Zahnfabrik), etc.
Answer: revised
- “The purpose of this study is to measure and compare color dimensions of zirconia”. Please specify the type of zirconia restoration (e.g., zirconia crowns).
Answer: LINE 19-20, The purpose of this study is to measure and compare color properties of zirconia crowns in different shades on ti-base abutments
- Study groups were not clarified in the abstract. Define the study groups before presenting their results.
Answer: LINE 23-25. Specimens were divided into 4 main groups depending zirconia shade (A1/0, A2/3, A3.5/4 and B2/3), then divided to 2 subgroups according to ti-base height.
- Please add a clinical take-home message to the conclusions in the abstract.
Answer: LINE 29-30, Using a 5.5-mm height ti-base abutment may produce a clinically unacceptable outcome ( DEab>2) in A1/0 and A2/3 color groups.
Introduction:
- Replace “Anterior fixed partial dentures (FDP)” with “anterior fixed prosthesis” or “anterior fixed prosthodontic restorations”.
Answer: revised.
- The following is not accurate: “screw-retained implant or cement-retained implant”. I believe this was a mistranslation, and the author meant “implant-supported screw-retained and cement-retained prostheses”.
Answer: revised
- Authors justified the reason of conducting this study.
Material and Methods:
- Please clarify how position was standardized while using Easyshade spectrophotometer probe. This is very important. A slight change in the position could result in different results. A clear illustrating figure could be very beneficial. Figure 2 did not show the special mold that was used to hold the Easyshade device clearly, although it was claimed to do so. Please explain how this was repeatable in all specimens.
Answer: the figure was added showing the mold for holding the device.
- In sample size calculations, authors claimed that effect size was 1.6. Please explain where this came from. Sample size calculations should be explained in more details.
Answer: LINE 248-346 Data analysis was performed using SPSS 27 for Windows (IBM SPSS Statistics, Chicago, IL). Descriptive analysis included mean, median and standard deviation. The Kolmogorov-Smirnov Test showed non normal distribution and E log transformation was made. Data were analyzed using 2-way ANOVA, and mean DE values were compared with t-test (alpha=.05 for all tests).
To test for a difference in mean between 2 groups, using the t-test, according to a sample size calculation that considers test significance = 0.05, power = 0.9, and effect size = 1.6,
(Effect size assumed a difference of 1.2 between the 2 groups with a pooled standard deviation of 0.75) as was performed in SPSS software, 9 samples in each group should be sufficient for achieving significant results at 0.05.
Results:
- Results are OK
Discussion:
- It is better to enrich the discussion section with more published results and comparisons.
Answer: the discussion section was revied thoroughly.
- Please justify why ti-base abutments were chosen for the present study.
Answer: At our university, only Straumann company has 2 abutment height, all other companies, have 1 abutment height. So for the study purposes, we used this type of ti-base.
- Cite publications that confirm the validity of using Easyshade.
Answer: publications were added.
- Chu, S.J.; Trushkowsky, R.D.; Paravina, R.D. Dental color matching instruments and systems. Review of clinical and research aspects. J. Dent. 2010, 38, e2-e16.
- Hardan, L.; Bourgi, R.; Cuevas-Suárez, C.E.; Lukomska-Szymanska, M.; Monjarás-Ávila, A.J.; Zarow, M.; Jakubowicz, N.; Jorquera, G.; Ashi, T.; Mancino, D.; Kharouf, N.; Haikel, Y. Novel Trends in Dental Color Match Using Different Shade Selection Methods: A Systematic Review and Meta-Analysis. Materials2022, 15, 468.
Conclusions:
- While drawing conclusions, kindly indicate that these were within the study limitations.
Answer: revised.
- Add clinical take-home messages to your conclusions
Answer: LINE 716-718 , 3. If possible, using a short ti-base abutment may be favorable when restoring a monolithic anterior zirconia A1/0 and A2/3 shade zirconia, however the retention forces must considered depending on the length of the restored crown.
Round 2
Reviewer 1 Report
Thank you for the arrangements and can be accepted.